# An Investigation of Scale-Resolving Turbulence Models for Supersonic Retropropulsion Flows

Gabriel Nastac and Abdelkader Frendi *

Department of Mechanical and Aerospace Engineering, University of Alabama in Huntsville, Huntsville, AL 35899, USA
* Correspondence: frendik@uah.edu

**Abstract:** Characterization of unsteady loads is critical for the development of control systems for next-generation air vehicles. Both Direct Numerical Simulation (DNS) and Large Eddy Simulation (LES) methods are prohibitively expensive, and existing Reynolds-Averaged Navier-Stokes (RANS) approaches have been shown to be inadequate in predicting both mean and unsteady loads. In recent years, scale-resolving methods, such as Partially Averaged Navier-Stokes (PANS) and Detached Eddy Simulation (DES), have been gaining acceptance and filling the gap between RANS and LES. In this study, we focus on a new variant of the PANS method, namely blended PANS or BPANS, which was shown to perform well in the incompressible regime for both wall-bounded and free shear flows. In this paper, we extend BPANS to compressible supersonic flows by adding a compressibility correction, leading to a new model called BPANS CC. The new model is tested using a well-known supersonic mixing layer case, and the results show good agreement with experimental data. The model is then used on a complex supersonic retropropulsion case and the results are in good agreement with experimental data.

**Keywords:** supersonic retropropulsion; turbulence; CFD

## 1. Introduction

Supersonic retropropulsion (SRP) flows important for various entry, descent, and landing (EDL) applications. As payloads become larger, conventional deceleration technologies such as parachutes become infeasible, necessitating alternate EDL strategies such as SRP. SRP is a key technology for re-usable rockets and proposed Human-scale Mars lander concepts [1]. Figure 1 depicts a characteristic flow field for SRP. A rocket engine plume expands out in front of a vehicle culminating into a terminal Mach disk. The Mach disk is a normal shock with incoming Mach numbers commonly over 10, leading to significant property variations over a small thickness. In addition, plume shear layers interact with the bow shock leading to a highly dynamic flow field with significant turbulence. Past experiments and simulations have shown that aerodynamic loads are not entirely insignificant and, thus, are important for vehicle design. Ultimately, unsteady load prediction is required to determine the necessary control authority of vehicles employing SRP.

While in reality, eventual SRP applications will involve high enthalpy chemically reacting flow, experimental limitations and complexities have led to an experimental focus on low-temperature perfect gas experiments to capture the general flow field behaviors. There have been numerous experiments exploring perfect gas SRP flows with corresponding simulations. For example, Ref. [2] simulates air experiments with various flow solvers using perfect gas assumptions and steady-state, 3-D Reynolds-averaged Navier-Stokes (RANS) with a reasonable agreement to each other and to the experimental data. Continuing work in Ref. [3] simulated more recent experiments [4] using unsteady methods, including detached-eddy simulation (DES) and unsteady RANS (URANS) using the same

three flow solvers with a generally favorable agreement to the experiments. Pre-test simulations of a human-scale Mars lander concept have also been investigated using URANS and DES approaches ([5,6]) based on upcoming perfect gas air experiments [7].

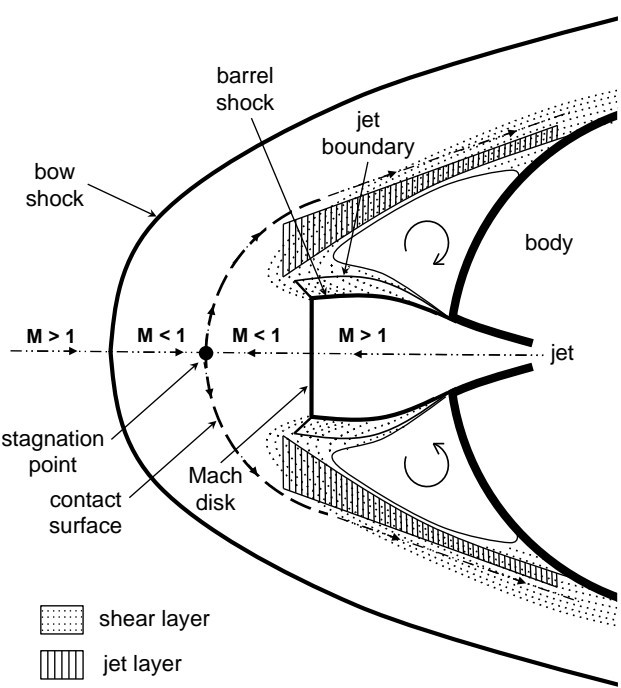

**Figure 1.** Characteristic flow field for supersonic retropropulsion (taken from [8]).

Partially Averaged Navier-Stokes (PANS) ([9,10]), a hybrid RANS-LES method, is considered to model these flows. PANS bridges RANS and Direct Numerical Simulation (DNS) through resolution control parameters ($f_k$, $f_\varepsilon$). One issue with URANS and DES approaches, is that in the limit of very fine grid resolution, the equations do not simplify to DNS. This contrasts with large-eddy simulation (LES) approaches, where the turbulence model is typically a function of the filter width, which reduces the model impact with finer filter (commonly tied to the grid size) resolution. PANS enables this bridge in a more fundamental fashion for hybrid RANS/LES approaches. PANS methods adjust various components in the RANS equations using unresolved-to-total ratios of turbulent kinetic energy, $f_k$, and dissipation, $f_\varepsilon$. When these ratios are unity, PANS equations become the RANS equations. As these ratios tend to zero, the original Navier-stokes equations are recovered, and you have DNS. The variation of turbulence modeling is smooth between these two limits. There are still closures necessary for PANS. In theory, these ratios are variable both in space and time for a given simulation. It is not possible to generally compute these variables a priori, especially for compressible flows typically encountered in aerospace. In practice, these ratios are constant throughout the flow field. While some closures do exist which can be used to estimate these ratios from an initial RANS simulation, they rely on incompressible homogeneous isotropic flow assumptions, which are not valid for all flow configurations. Another approximation commonly employed for high Reynolds number flows, which is used in this work, is that the dissipation scales are assumed to not be resolved at all, thus leading to a dissipation ratio of unity. The only control parameter is thus the ratio of underresolved-to-total turbulent kinetic energy.

In this paper, we investigate the use of a recently developed Blended PANS [11] or BPANS model. This model uses the advantages of a $k - \omega$ based PANS near viscous walls and a $k - \varepsilon$ based PANS in the freestream. The blending follows the method developed by Menter's baseline model [12]. BPANS was shown to have superior performance for incompressible flows over a backward-facing step and a circular cylinder [11]. The following work investigates the extension of BPANS to compressible flows. To this end, a

compressibility correction will be added to BPANS to accommodate these flows leading to the new BPANS CC model.

The paper outline is as follows. First, the compressible flow governing equations are described, including a compressible BPANS model variant with associated compressibility corrections. The numerical implementation is then described. Simulations of a supersonic mixing layer are performed to demonstrate the applicability of the method to supersonic flows. Simulations of an experimental SRP setup are then performed using URANS, DES, BPANS, and BPANS CC and compared against experimental data.

## 2. Materials and Methods

### 2.1. Governing Equations

The governing equations for a turbulent compressible perfect gas flow are the conservation of mass, momentum, and total energy

$$\frac{\partial}{\partial t}\left(\overline{\rho}\right) + \frac{\partial}{\partial x_j}\left(\overline{\rho}\widetilde{u}_j\right) = 0, \tag{1}$$

$$\frac{\partial}{\partial t}\left(\overline{\rho}\widetilde{u}_i\right) + \frac{\partial}{\partial x_j}\left(\overline{\rho}\widetilde{u}_i\widetilde{u}_j + \overline{p}\delta_{ij}\right) - \frac{\partial}{\partial x_j}\left(\overline{\tau}_{ij}\right) = 0, \tag{2}$$

$$\frac{\partial}{\partial t}\left(\overline{\rho}\widetilde{E}\right) + \frac{\partial}{\partial x_j}\left(\left(\overline{\rho}\widetilde{E} + \overline{p}\right)\widetilde{u}_j\right) - \frac{\partial}{\partial x_j}\left(\widetilde{u}_k\overline{\tau}_{kj} + \overline{q}_j + \left(\widetilde{\mu} + \sigma_k\mu_t\right)\frac{\partial k}{\partial x_j}\right) = 0, \tag{3}$$

where Reynolds-averaged and Favre-averaged variables are denoted by $\overline{(\cdot)}$ and $\widetilde{(\cdot)}$ and respectively. $\overline{\rho}$ is the mixture density, $\widetilde{u}_i$ is the $i$th component of velocity, and $\widetilde{E}$ is the total energy. $\overline{p}$ is the pressure, $\overline{\tau}_{ij}$ is the shear stress tensor, and $\overline{q}_j$ is the $j$th component of the heat flux.

Constitutive relations for pressure, energy, shear stress tensor, and heat transfer are required to close the equation set. The gas is assumed to be an ideal gas; the pressure is thus defined as

$$\overline{p} = \overline{\rho}R\widetilde{T} \tag{4}$$

where $\widetilde{T}$ is the temperature, and $R$ is the gas constant of air.

The total energy $\widetilde{E}$ is defined as

$$\widetilde{E} = C_v\widetilde{T} + \frac{1}{2}\widetilde{u}_k\widetilde{u}_k + k \tag{5}$$

where $C_v$ is the specific heat at constant volume. The specific heat ratio, $\gamma = C_p/C_v$, where $C_p$ is the specific heat at constant pressure, is assumed constant and equal to 1.4.

The turbulent kinetic energy (TKE), $k$, is defined as

$$k = \frac{1}{2}\widetilde{u_k''u_k''}. \tag{6}$$

Viscous transport is closed with a Newtonian model with turbulence modeled using the Boussinesq eddy viscosity assumption [13]

$$\overline{\tau}_{ij} = 2(\widetilde{\mu} + \mu_t)\,\overline{S}_{ij} - \frac{2}{3}\overline{\rho}k\delta_{ij} \tag{7}$$

where $\widetilde{\mu}$ is the dynamic viscosity, $\mu_t$ is the turbulent eddy viscosity computed by a turbulence model, and $\overline{S}_{ij}$ is the strain rate tensor computed as

$$\overline{S}_{ij} = \frac{1}{2}\left(\frac{\partial \widetilde{u}_i}{\partial x_j} + \frac{\partial \widetilde{u}_j}{\partial x_i}\right) - \frac{1}{3}\frac{\partial \widetilde{u}_k}{\partial x_k}\delta_{ij}. \tag{8}$$

Heat transfer is closed with Fourier's law

$$\bar{q}_j = (\widetilde{\kappa} + \kappa_t)\frac{\partial \widetilde{T}}{\partial x_j}, \tag{9}$$

where $\widetilde{\kappa}$ is the thermal conductivity and $\kappa_t = C_p \mu_t / Pr_t$ is the turbulent contribution to thermal conductivity, where $Pr_t$ is the turbulent Prandtl number, which is assumed constant. The transport properties (viscosity and thermal conductivity) are computed using Sutherland's law and a constant Prandtl number of 0.71.

Turbulence is modeled using DES and PANS methods. For DES [14], the one-equation Spalart-Allmaras model [15] with Catris-Aupoix compressibility corrections [16] is employed. The conservation form of the PANS model is used for this work. The turbulent kinetic energy is coupled to the total energy equation as written in Equation (5), contributes to the turbulent stress, and the turbulence model also includes compressibility corrections. The BPANS transport equations are given as

$$\frac{\partial}{\partial t}(\bar{\rho}k) + \frac{\partial}{\partial x_j}(\bar{\rho}k\widetilde{u}_j) - \frac{\partial}{\partial x_j}\left((\widetilde{\mu} + \sigma_k\mu_t)\frac{\partial k}{\partial x_j}\right) = S_k, \tag{10}$$

$$\frac{\partial}{\partial t}(\bar{\rho}\omega) + \frac{\partial}{\partial x_j}(\bar{\rho}\omega\widetilde{u}_j) - \frac{\partial}{\partial x_j}\left((\widetilde{\mu} + \sigma_\omega\mu_t)\frac{\partial \omega}{\partial x_j}\right) = S_\omega. \tag{11}$$

The source terms are

$$S_k = \min(P, 20\beta^*\bar{\rho}\omega k) - \beta^*\bar{\rho}\omega k + S_k^{CC}, \tag{12}$$

$$S_\omega = \frac{\bar{\rho}\gamma}{\mu_t}P - \beta\,\bar{\rho}\omega^2 + 2(1 - F_1)\frac{\bar{\rho}\sigma_{\omega 2}}{\omega}\frac{\partial k}{\partial x_j}\frac{\partial \omega}{\partial x_j} + S_\omega^{CC}, \tag{13}$$

and the remaining auxiliary functions are

$$P = \bar{\tau}_{ij}^{turb}\frac{\partial \widetilde{u}_i}{\partial x_j}, \tag{14}$$

$$\bar{\tau}_{ij}^{turb} = 2\mu_t\overline{S}_{ij} - \frac{2}{3}\bar{\rho}k\delta_{ij}, \tag{15}$$

$$\mu_t = \frac{\bar{\rho}k}{\omega}, \tag{16}$$

$$F_1 = \tanh\left(arg_1^4\right) \tag{17}$$

$$arg_1 = \min\left(\max\left(\frac{\sqrt{k}}{\beta^*\omega d}, \frac{500\widetilde{\mu}}{\bar{\rho}d^2\omega}\right), \frac{4\bar{\rho}\sigma_{\omega 2}k}{CD_{k\omega}d^2}\right), \tag{18}$$

$$CD_{k\omega} = \max\left(2\bar{\rho}\sigma_{\omega 2}\frac{1}{\omega}\frac{\partial k}{\partial x_j}\frac{\partial \omega}{\partial x_j}, 10^{-20}\right). \tag{19}$$

The model constants are a blend of inner (1) and outer (2) constants $\phi = F_1\phi_1 + (1 - F_1)\phi_2$. The BPANS-adjusted constants for the model, include the ratios of underresolved-to-total turbulent kinetic energy, $f_k$, and dissipation, $f_\varepsilon$, and are computed with the following:

$$\beta^* = 0.09, \kappa = 0.41, \tag{20}$$

$$\gamma_1 = \frac{5}{9}, \ \gamma_2 = 0.42, \ \sigma_{\omega 1} = 0.5\frac{f_\varepsilon}{f_k^2}, \sigma_{\omega 2} = \frac{1}{1.3}\frac{f_\varepsilon}{f_k^2}, \tag{21}$$

$$\sigma_{k1} = 0.5\frac{f_\varepsilon}{f_k^2}, \sigma_{k2} = \frac{f_\varepsilon}{f_k^2}, \tag{22}$$

$$\beta_1 = 0.05\left(1 - \frac{f_k}{f_\varepsilon}\right) + 0.075\frac{f_k}{f_\varepsilon}, \tag{23}$$

$$\beta_2 = 0.0378 + \frac{f_k}{f_\varepsilon}0.045. \tag{24}$$

The compressibility correction (CC) sources utilize the Suzen and Hoffman compressibility correction [17] experimentally fit to match turbulent compressible mixing data. Two additional sources are added to the total dissipation: an additional dissipation due to compressibility effects and an additional term to incorporate additional dissipation due to pressure dilatation. Using the $k - \varepsilon$ model, the new terms are:

$$\varepsilon_C = \alpha_1\bar{\rho}\varepsilon M_t^2 \tag{25}$$

$$\overline{p''d''} = -\alpha_2 P_k M_t^2 + \alpha_3\bar{\rho}\varepsilon M_t^2. \tag{26}$$

All the terms include a turbulent Mach-squared dependence due to experimental data available. For BPANS, no additional adjustments are necessary; all the terms match the general dissipation of TKE, which is unchanged for BPANS formulations. Transforming these source terms to $k - \omega$ models ($\varepsilon = \beta^* k\omega$) leads to the following:

$$S_k^{CC} = (1 - F_1)\left(-\alpha_1\bar{\rho}\beta^* k\omega M_t^2 + \overline{p''d''}\right), \tag{27}$$

$$S_\omega^{CC} = (1 - F_1)\left(\alpha_1\beta^*\bar{\rho}\omega^2 M_t^2 - \frac{\overline{\rho}}{\mu_t}\overline{p''d''}\right) \tag{28}$$

where the turbulent Mach number is $M_t = \sqrt{2k/a^2}$, and $a$ is the local speed of sound. The pressure dilatation coefficients are fit by Sarkar (see [16]) based on DNS data ($Re_\lambda \approx 25$) and the additional dissipation constants are fit on experimental data: $\alpha_1 = 1.0$, $\alpha_2 = 0.4, \alpha_3 = 0.2$. These source terms are only enabled away from walls through the $F_1$ function due to known issues with compressibility corrections underpredicting skin friction for high-speed turbulent boundary layers, which can impact separation [18].

*2.2. Numerical Implementation*

Fully Unstructured 3D Navier-Stokes (FUN3D) is employed as the CFD solver for this work. FUN3D(Version 13.7-b3b47a4, NASA Langley Research Center, Hampton, VA, 2022) is a CFD software developed at the NASA Langley Research Center and is used to simulate and analyze problems across the speed range from incompressible flows to hypersonic flows [19]. The equations are solved with the Method of Lines (MOL). The spatial domain is discretized using a node-based finite volume approach on general unstructured grids. The degrees of freedom are stored in the nodes of the grid. A dual-grid system is used; the primal grid is composed of tetrahedrons, pyramids, prisms, and hexahedrons. A dual grid is generated by bisecting every edge to generate polyhedrons at each node. The equations are integrated in time implicitly for this work using a second order backward-difference (BDF2) method. The inviscid fluxes and analytical Jacobians are computed at each dual face of the grid using HLLE++ [20], an approximate Riemann solver. Second-order accuracy is obtained using MUSCL (Monotonic Upstream-centered Scheme for Conservation Laws) with unweighted least-squares gradients computed at each cell. In this work, the inviscid fluxes are computed as a blend of central fluxes and upwind fluxes to reduce numerical dissipation. The shock sensor of the HLLE++ scheme is used as the sensor to toggle the blending; upwind fluxes are used in areas of shocks, and a 90% central flux blend is used elsewhere for stability. The viscous fluxes and analytical Jacobians are computed using a Galerkin-based approach over the primal cells of the domain and distributed to the nodes. The mean flow and turbulence equations are solved in a fully coupled fashion.

## 3. Results

A supersonic mixing layer is first simulated to demonstrate BPANS CC on a canonical flow. Supersonic retropropulsion flows are then simulated to demonstrate the BPANS CC approach with comparisons to URANS and DES methods as well as experimental data.

### 3.1. Supersonic Mixing Layer

High-speed spatially developing shear layer simulations of the Goebel and Dutton experiments [21] have been carried out using BPANS and BPANS CC. The specific condition investigated is case one which has a freestream Reynolds number per meter based on the upper condition of about $Re_\infty = 30 \times 10^6$ and the two mixing stream Mach numbers are $M_1 = 2.01$ and $M_2 = 1.38$, respectively. A 45 million cells hexahedral multi-block structured grid was generated for a domain of length 0.35 meters which include the upstream splitter plate, which is modeled with a length of 0.05 m. The splitter plate thickness is 500 microns. The simulation width is 0.01 m which is discretized into 32 uniform-width cells. The walls are resolved and modeled with no slip. The inflows are set to supersonic conditions corresponding to the experimental conditions. Extrapolation is utilized on the outflow plane as the flow is supersonic. The two side planes are assumed periodic, and the top and bottom domains are modeled as z-symmetry. The walls have meshed with a $y^+ = 1 \approx 1$ µm based on the freestream Reynolds number and cells with x-y aspect ratio of 4 and $O(y^+ \approx 100)$ were generated in the bulk of the shear layer. The width spacing, $z^+$, is $O(y^+ \approx 300)$. The x-y aspect ratio of the cells at the plate lip is unity. The equations are integrated in time using a two-step backward difference (BDF2) scheme with a time step of 0.25 µs which corresponds to a global CFL of about 80 due to resolving the wall and is of order unity in the shear layer away from the plate. Five sub iterations are used, which correspond to nominally 2 orders of magnitude residual reduction for the equations. Once statistically stationary flow is achieved, statistics are obtained over 5 flows through times, where a flow through time is defined by the top plate freestream velocity and domain length.

BPANS is run for $f_k$ ratios of unity and 0.2, the latter implying the grid is resolving 80% of the turbulent kinetic energy. Compressibility corrections are also tested for both ratios. The stream velocity similarity profile is shown in Figure 2. The mixing layer thickness, $b$, is defined as the transverse distance between mean streamwise velocities of $U_1 - 0.1\Delta U$ and $U_2 + 0.1\Delta U$. Results are found to be self-similar and free from lip shock effects starting from x = 0.10 m. The growth rate is obtained from x = 0.10 m and x = 0.25 m locations. Figure 3 depicts non-dimensional y-velocity (periodic direction) contours. RANS results ($f_k = 1.0$) predict a sharper mixing layer curve than the experimental data, as has been commonly shown in past studies in the literature. The RANS results do not become unsteady and thus have no y-velocity components. Both BPANS approaches are unsteady and better predict the mixing layer curve versus the RANS results. For both ratios, the compressibility-corrected models better predict the experimental growth rate ($db/dx$). The predicted growth rate error is reduced from 15% to less than 5% with the compressibility correction for this condition for both $f_k = 1.0$ and $f_k = 0.2$. Overall, the results indicate that BPANS CC can be used to successfully predict canonical supersonic compressible flows. The $f_k = 0.2$ results matching experimental mixing and growth rate demonstrate the capability to simulate unsteady flows, which are increasingly becoming important for the prediction of unsteady loads for vehicle design and analysis.

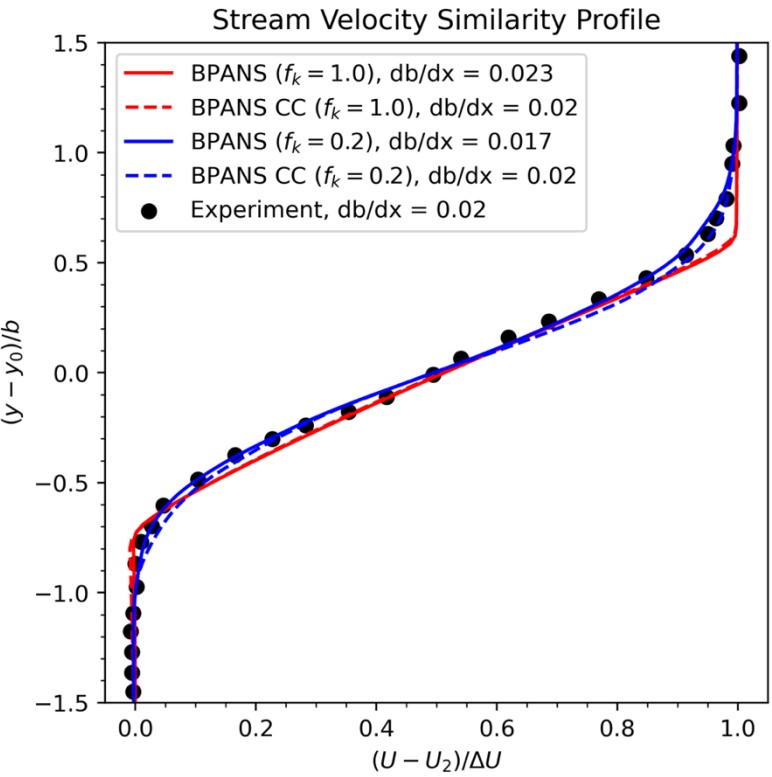

**Figure 2.** Stream velocity similarity profiles with comparison to experimental data [21].

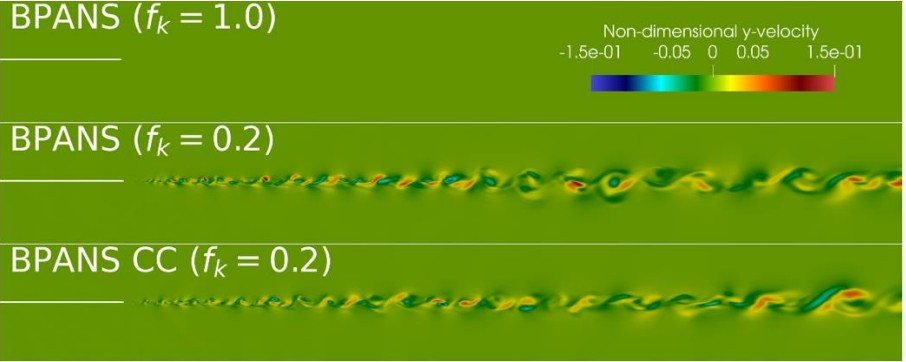

**Figure 3.** Mixing layer y-velocity contours at the y-centerline. The RANS results ($f_k = 1.0$) do not exhibit unsteadiness and thus have a y-velocity of zero. The velocity is nondimensionalized by the upper plate velocity.

### 3.2. Supersonic Retropropulsion Flows

A perfect gas single-nozzle experimental supersonic retropropulsion flow configuration [4] is investigated with various turbulence modeling approaches. The freestream Mach number is 4.6, and the freestream Reynolds number per meter is 5 million. The heat shield is a 70-degree sphere cone with a diameter of 0.127 m. The engine exit diameter, $D_e$, is 0.0127 m. The nozzle exit-to-throat area ratio is 4. The plenum total pressure and temperature are set according to the experimental setup as $\frac{p_0}{p_\infty} = 7724.3$ and $\frac{T_0}{T_\infty} = 5.34$. The freestream air is cold, and thus a perfect gas assumption is valid and used for these simulations.

Unstructured grids are utilized here due to the complexity of the geometry. A family of unstructured grids is generated with varying refinement. Figure 4 depicts the first unstructured grid consisting of 80 million cells and 15 million points. All walls are modeled as no-slip adiabatic walls. A prismatic boundary layer with an initial wall spacing targeting

a $y^+ = 1$ is generated, with the farfield grid composed of tetrahedra. There is an engine spacing source of $\Delta / D_e = 0.04$. A spherical source outside the nozzle exits and surface has a spacing of $\Delta / D_e = 0.15$. Farfield spacing is set to $\Delta / D_e = 0.4$. Two finer grids are generated by decreasing the surface and volume spacing sources by 25% and 50%, leading to grids approximately two and three times larger than the original grid. Steady-state BPANS ($f_k = 1.0$), or RANS, simulations are performed on the three grids, and results are presented in Figure 5. Surface pressure coefficient data is plotted along the vehicle with experimental comparisons with error estimates. All three grids predict nearly indistinguishable surface property results. Grid convergence is obtained successfully, with the original 80 million cell grid being adequately resolved for surface property prediction, which are the engineering quantities of interest.

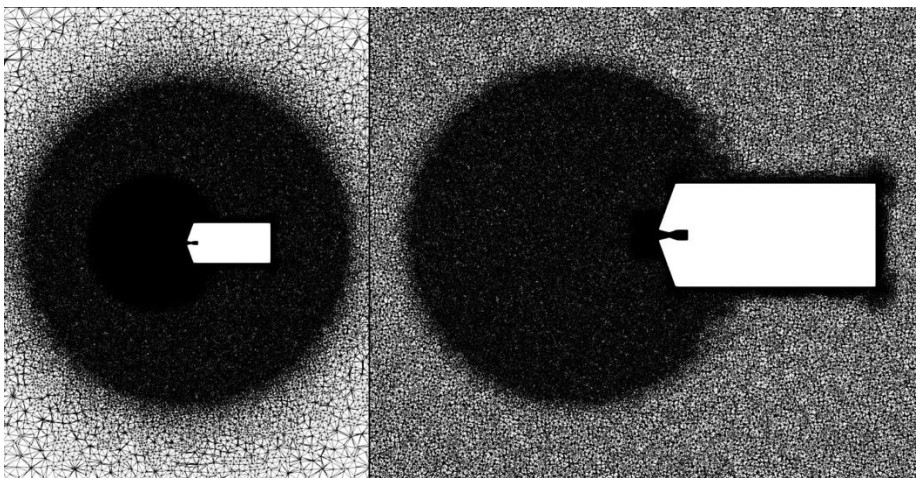

**Figure 4.** Y-plane centerline slice of 80 million cell SRP unstructured grid.

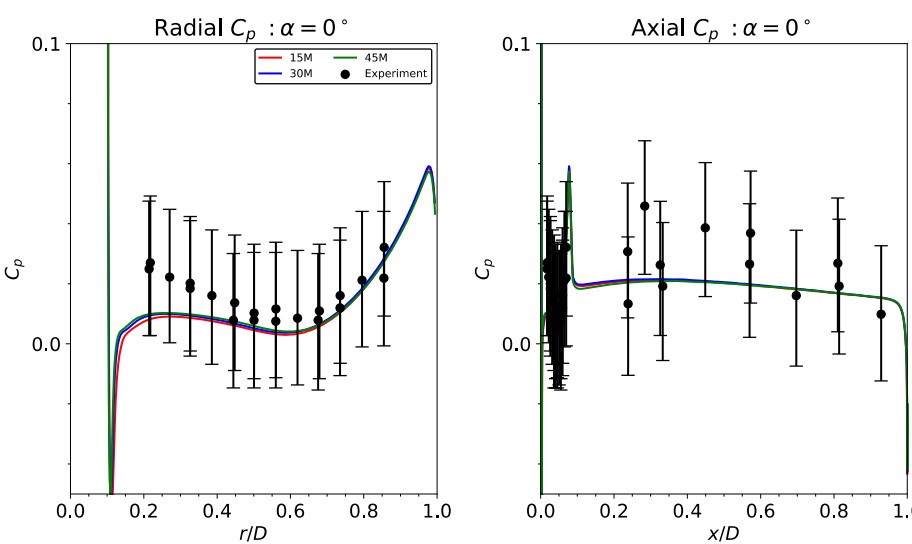

**Figure 5.** SRP grid convergence study using RANS.

BPANS, BPANS CC, and DES turbulence models are used for unsteady flow prediction. Equations are integrated in time with BDF2 with a time step of one microsecond. Five subiterations are used, leading to nominally two orders of magnitude residual reduction. Skew-symmetric blended inviscid fluxes are utilized. BPANS and BPANS CC with $f_k = 0.2$ are used. Three angles of attack are considered: $0°, 12°$, and $20°$. Simulations are time-averaged after quasi steady state is obtained for roughly 10 milliseconds, or about 20 periodic cycles for the $0°$ angle of attack configuration.

The nominal thrust coefficient is $C_T = 2.0$, with all models predicting this coefficient within 2% of the nominal. For this example, the thrust is the dominant contribution to the total force; the maximum aerodynamic component is roughly 15% of the total. While aerodynamic contributions are not a significant component of the mean forces, aerodynamic fluctuations are fundamentally what drives what the vehicle guidance navigation and control systems are required to provide for stability and control.

Figure 6 depicts surface pressure coefficient data. Mean pressure coefficients agree well with experimental data. The models predict unsteady flowfields, which contrasts with the RANS approach (Figure 5), which did not have significant variance. BPANS with $f_k = 0.2$ better predicts pressure coefficient data compared to $f_k = 1.0$ (Figure 5), which emphasizes the improved prediction capabilities of unsteady hybrid RANS/LES models versus traditional RANS approaches. The BPANS CC model better predicts nose surface pressure for the 12° and 20° angle of attack cases. Along the axial part of the vehicle, the impact of the compressibility correction is small since flow Mach numbers are much smaller versus the jet plume at the forebody.

Overall, BPANS CC with $f_k = 0.2$, and DES models compare very favorably. For the larger angle of attack, BPANS CC better predicts the surface pressure on the nose of the vehicle versus DES. Standard deviations are shown for the zero angles of attack configuration in Figure 7. Standard deviations are overpredicted slightly but follow experimental trends for all the unsteady models. The largest discrepancy in standard deviation occurs at the mid-radial point. The experiment considered random error, flowfield nonuniformity, and model/instrumentation asymmetries for the error bars, with most of the error due to flowfield nonuniformity. The simulations here neglect freestream turbulence fluctuations which can impact shock dynamics and, ultimately, surface property prediction. In addition, side walls are not modeled; side wall turbulence does exist for this facility which can be seen in the numerical schlieren of the experiments. Aerodynamic drag for the three angles of attack is shown in Figure 8. The forces on the vehicle are nearly periodic for the zero angles of attack configuration and become more unsteady and ultimately chaotic for the higher angles of attack. The force frequencies decrease as the angle of attack increases as well.

Numerical schlieren is shown in Figure 9 in comparison to experimental schlieren [4]. As previously mentioned, the wind tunnel is not modeled, and thus, boundary layer turbulence and, consequently, density fluctuations in the farfield are not present in the simulations. In addition, the numerical schlieren generated here is centerline slices based on density gradient magnitudes and not a volume integration of the whole flowfield and schlieren direction. Nonetheless, the numerical and experimental schlieren match extremely well. SRP flow features are characterized by the experimental measurements, including jet plume length ($L_J$), bow shock stand-off distance ($L_S$), maximum jet plume radius ($R_J$), and bow shock radius ($R_S$). Figure 10 depicts these quantities visually and Table 1 presents current comparisons using BPANS CC against experimental results obtained from schlieren. Flow features match very well, with the simulated key quantities of interest all within 3% of experimental data. Isosurfaces of the Q-criterion are presented for the three angles of attack in Figure 11. The increased turbulence and loss of symmetry at higher angles of attack are clearly visible in the isosurfaces and follow the vehicle drag trends plotted in Figure 8.

**Table 1.** SRP geometrical flow features compared to experimental data [4].

| Case/Flow Feature | $L_s$ [$m$] | $R_s$ [$m$] | $L_J$ [$m$] | $R_J$ [$m$] |
|---|---|---|---|---|
| Experiment | 0.183 | 0.246 | 0.129 | 0.077 |
| BPANS CC ($f_k = 0.2$) | 0.182 | 0.245 | 0.126 | 0.076 |

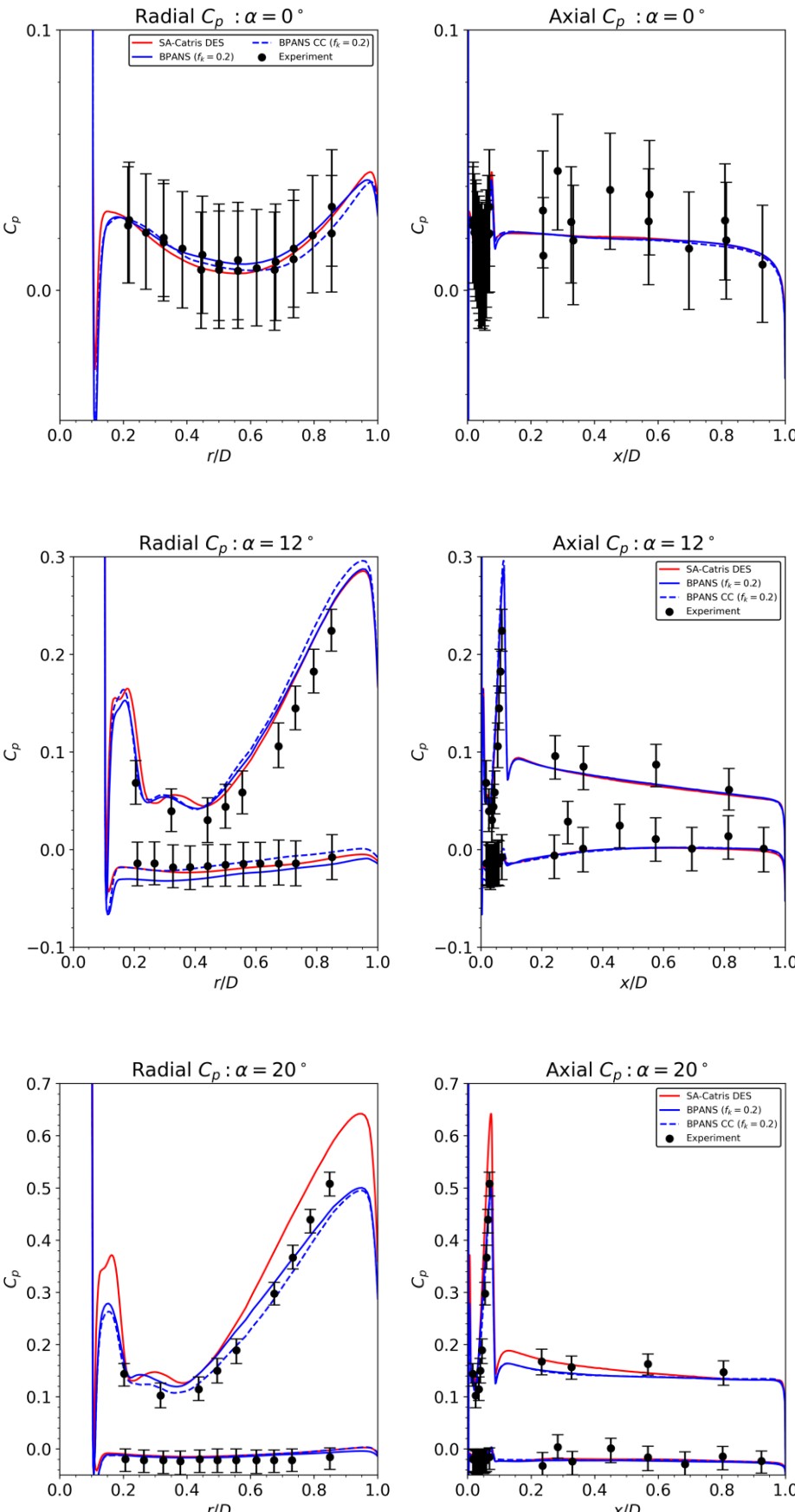

**Figure 6.** Surface pressure coefficient statistics for the SRP configuration.

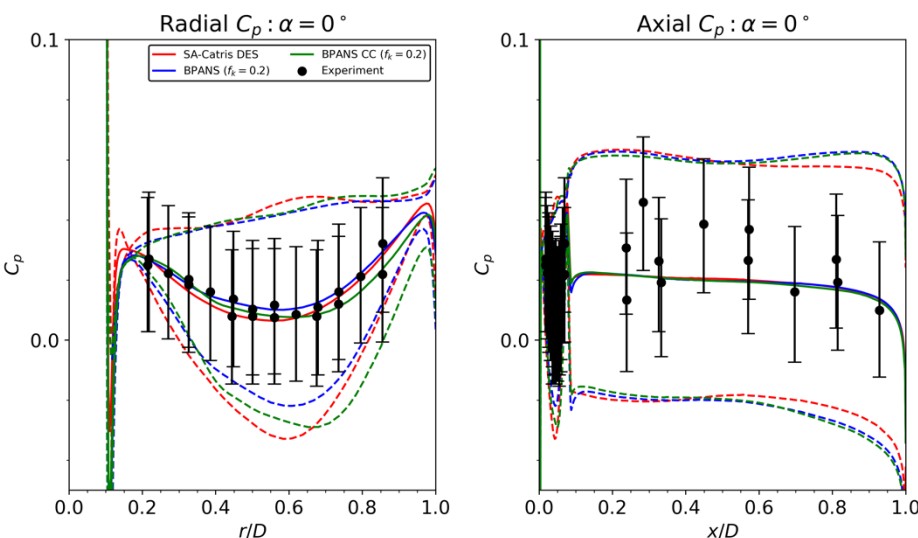

**Figure 7.** Surface pressure coefficient statistics with variance for the SRP configuration.

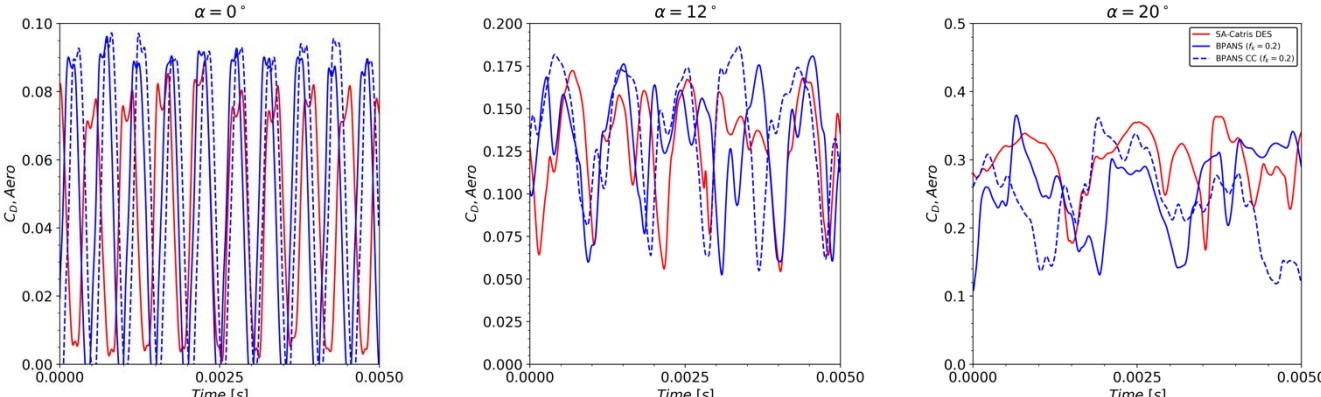

**Figure 8.** Aerodynamic drag versus time for the different angles of attack.

The turbulent kinetic energy spectrum at one vertical radius (near the transition lip) from the nozzle exit is plotted in Figure 12 using BPANS CC results for the $0^\circ$ the angle of attack case. An incompressible assumption is employed for the spectrum computation in which density fluctuations are neglected since a statistically stationary state does exist for this flowfield. The inertial subrange slope follows Kolmogorov's hypothesis for incompressible flows. The overall sound pressure level (OASPL) on the surface for the three angles of attack is plotted in Figure 13. The $0^\circ$ the angle of the attack case is nearly concentric, as expected, and has the lowest OASPL with a mean of around 140 dB on the nose. The $12^\circ$ the angle of attack case has a larger OASPL at around 155 dB, with the highest angle of attack having the highest OASPL at around 160 dB, both maximums occurring on the windward side and the impingement point downstream. Pressure spectra are presented in Figure 14. Pressure tap data at experimental probe locations are recorded every 20-time steps to match the experimental recording rate of 20 μs. Both SA-Catris and BPANS CC spectra results are computed. The models are comparable, with SA-Catris predicting slightly lower dominant frequencies than BPANS CC. Both models predict two peak frequencies. The CFD is only run for 10 ms, whereas the experiment is run for many seconds, so the vast difference in temporal scales must be considered when comparing results. In addition, CFD time steps must be small enough to capture higher frequencies, hence the absence of the higher frequency peaks, 6.6 kHz and 8.8 kHz, shown on the experimental spectrum from the CFD results. In terms of dominant frequency prediction for the $0^\circ$ angle of attack case, BPANS CC best predicts the experimental frequency of 2.2 kHz at 2.0 kHz, with SA-Catris DES predictions at 1.9 kHz. There are also still uncer-

tainties regarding the impact of the experimental wind tunnel geometry on these frequency results (e.g., the impact of freestream conditions and turbulence on the shock dynamics). Overall, the results compare favorably to past simulations in the literature ([3,22]).

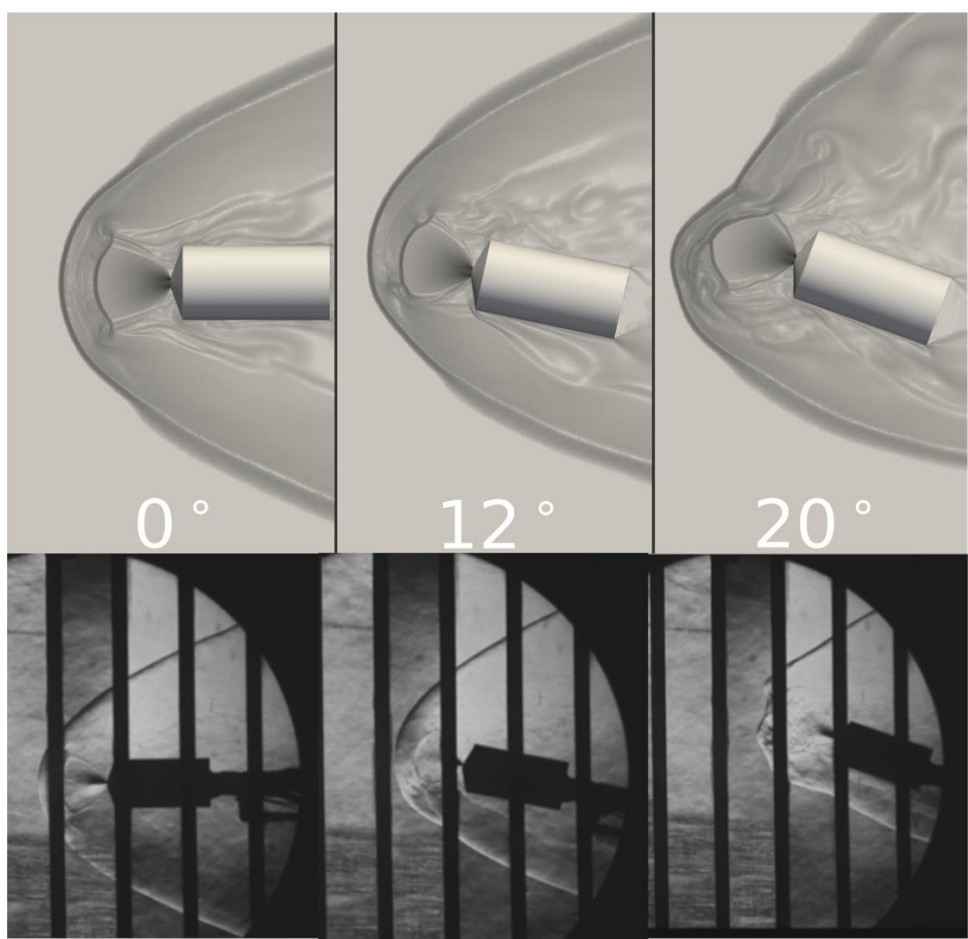

**Figure 9.** (**Top**): Log of density gradient centerline contours for BPANS CC model. (**Bottom**): Experimental schlieren [4].

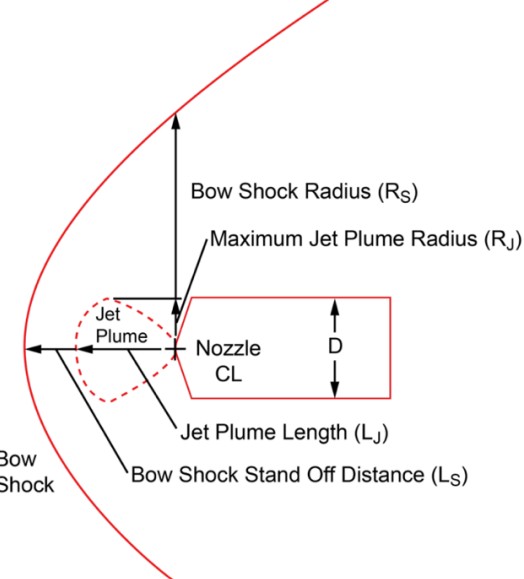

**Figure 10.** SRP geometrical flow feature schematic for the single-nozzle configuration [4].

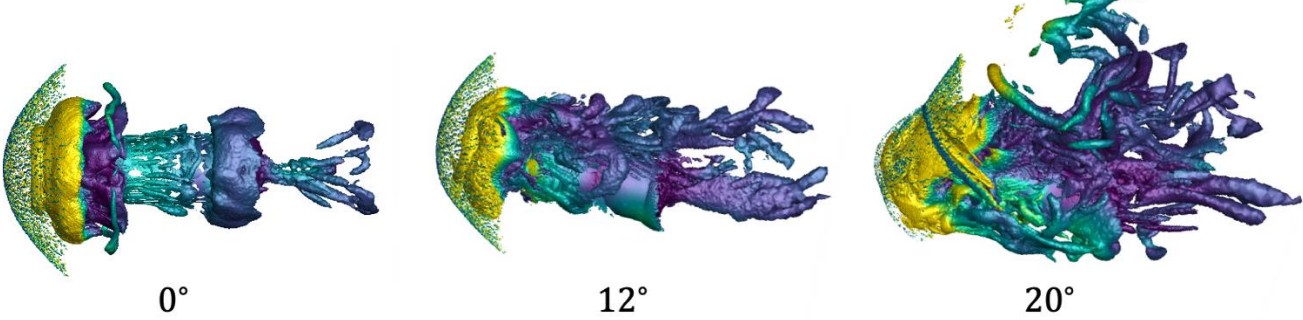

**Figure 11.** Q-criterion isosurfaces (15,000 per second) for the three angles of attack for BPANS CC ($f_k = 0.2$) colored by pressure.

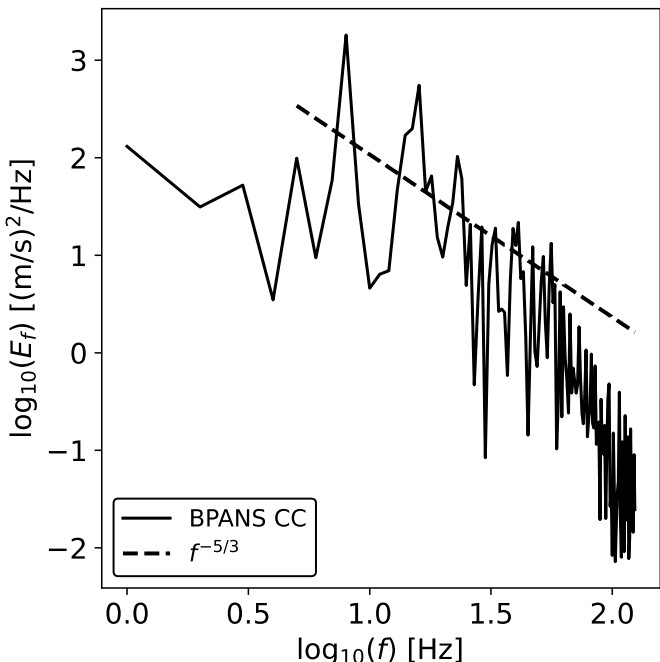

**Figure 12.** Turbulent kinetic energy spectrum at y/R = 1.

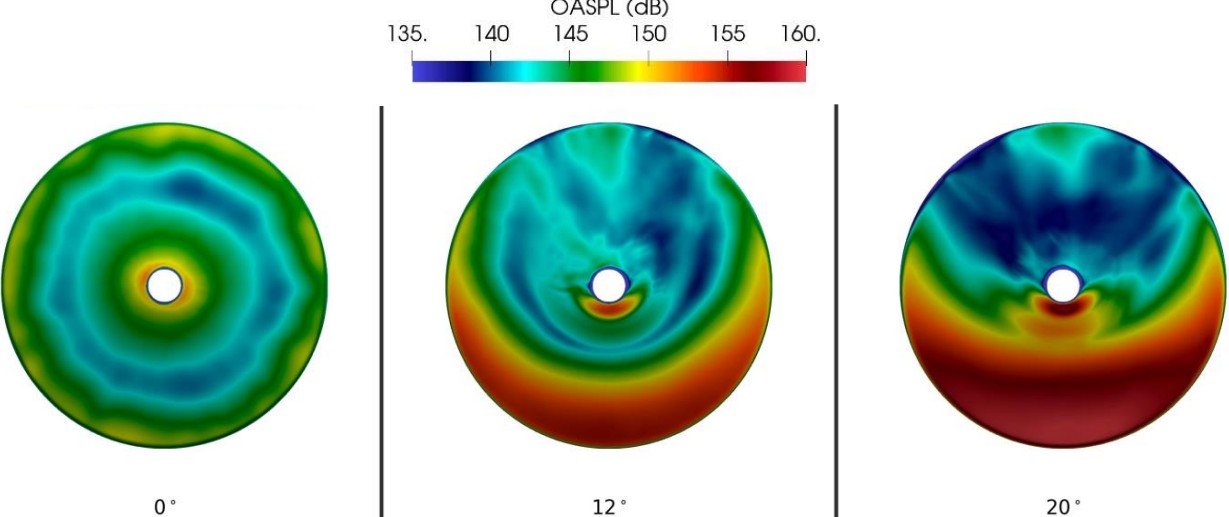

**Figure 13.** Overall Sound Pressure Level (OASPL) in decibels on the nose of the vehicle for the three angles of attack for BPANS CC ($f_k = 0.2$).

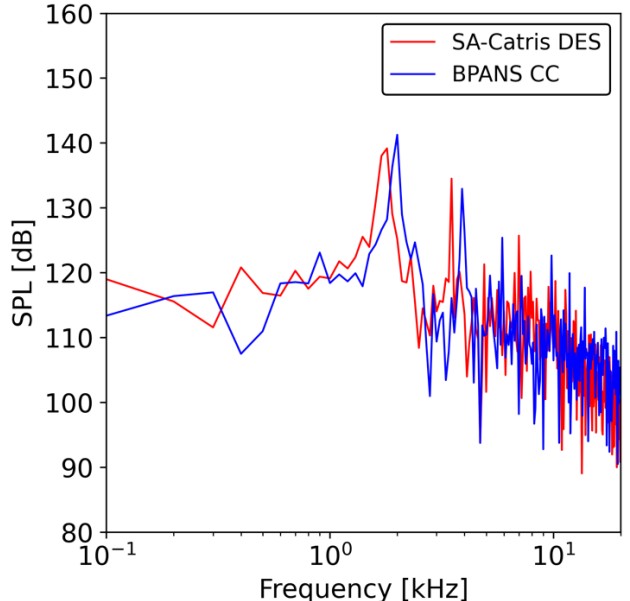
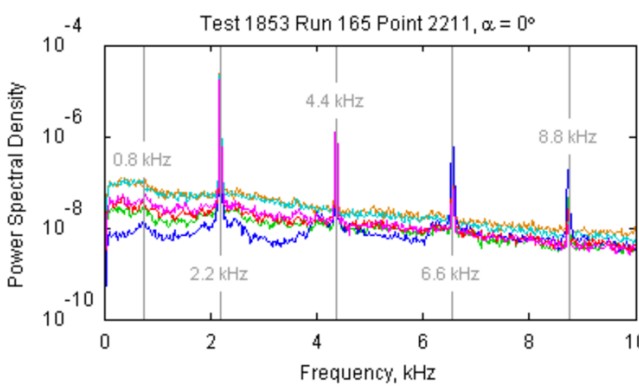

**Figure 14.** Pressure spectra plot for BPANS CC ($f_k = 0.2$) and SA-Catris DES compared to experimental data [23].

## 4. Conclusions

The scale resolving BPANS turbulence model has been extended to compressible flows leading to a new model called BPANS CC. The new model has been demonstrated on both a canonical compressible mixing layer flow and an SRP configuration with successful comparisons to experimental data. The model is capable of matching supersonic mixing layer mixing curves and growth rates.

For the complex SRP configuration, BPANS CC adequately predicts experimental surface data and geometrical flow features. It is found that compressibility corrections improved predictions over the non-corrected cases. The corrections led to improved surface pressure predictions on the heat shield of the vehicle at higher angles of attack.

The combination of BPANS and BPANS CC provides an opportunity to simulate turbulent flows from low subsonic to hypersonic speeds at low computational cost and reasonable accuracy.

**Author Contributions:** Conceptualization, A.F. and G.N.; methodology, G.N.; software, G.N.; validation, G.N.; formal analysis, G.N.; investigation, G.N.; resources, G.N.; data curation, G.N.; writing—original draft preparation, G.N.; writing—review and editing, A.F.; visualization, G.N.; supervision, A.F.; project administration, A.F. All authors have read and agreed to the published version of the manuscript.

**Funding:** This research received no external funding.

**Data Availability Statement:** Not applicable.

**Conflicts of Interest:** The authors declare no conflict of interest.

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
