# Peer review of "An Investigation of Scale-Resolving Turbulence Models for Supersonic Retropropulsion Flows"

_fluids, doi:10.3390/fluids7120362_

Round 1

Reviewer 1 Report

My comments I put them into attached file.

Reviewer 2 Report

In this work, the authors extend the blended partially-averaged Navier-Stokes (BPANS) to compressible flows and test the developed methodology on a mixing layers case and for supersonic retro-propulsion flow. Overall the article is written well. I have the following suggestions/comments for improving the quality of the manuscript.

1. The abstract seems too generic. For e.g. till line 16 of the abstract it is quite generic and it may please be re-written such that the main points are brought about.

2. Several acronyms such as BPANSCC etc are used without defining them at the first instant. Please check.

3. In Eqns. 2 and 3 both lower case and upper case indices are used? It is not clear why this is done. e.g i, j as well as \tau_{IJ}...please check the entire manuscript. Also in Equation 7.

4. It is not clear why a constant value of Pr = 0.7 is used? Given that the flow could be compressible would the Pr number be the same at all flow speeds?

5. The acronmyms FUN3D, HLLE, MUSCL could be defined at first use or in list of symbols and acronyms.

6. In line number 226, besides the Re value what does "1/m" represent?

7. For the supersonic retro-propulsion test case, the 80 million grid seems too high, could this be compared with other simulations existing in the literature?

8. To test the developed methodology, a simple case such as mixing layer was used? Why not consider a case that has walls for a better validation.

Reviewer 3 Report

The authors presented a very interesting Investigation of Scale-Resolving turbulence Models for Supersonic Retropropulsion Flows.

The paper is well prepared with high scientific soundness.

More details on the numerical method are to be provided.

Ref. [20] used for the validation is to be described.
